# Cretan Dittany (*Origanum dictamnus* L.), a Valuable Local Endemic Plant: In Vitro Regeneration Potential of Different Type of Explants for Conservation and Sustainable Exploitation

**DOI:** 10.3390/plants12010182

**Published:** 2023-01-01

**Authors:** Virginia Sarropoulou, Eleni Maloupa, Katerina Grigoriadou

**Affiliations:** Hellenic Agricultural Organization (HAO)-DIMITRA, Institute of Plant Breeding and Genetic Resources, Balkan Botanic Garden of Kroussia, Thermi, 57001 Thessaloniki, Greece

**Keywords:** aromatic–medicinal plant, basal culture media, Cretan dittany, explant types, ex situ conservation, Greek flora, in vitro organogenesis, plant growth regulators

## Abstract

*Origanum dictamnus* L. is a medicinal local endemic to the Island of Crete, Greece. Its propagation through biotechnological tissue culture techniques is essential due to its augmented multi-industrial sector demand. For direct organogenesis, among different culture media variants (MS, Gamborg B5), and cytokinins [6-benzyladenine (BA), kinetin (Kin), 2-isopentenyl adenine (2-iP)], the MS + added with BA (2.2 μM) was the most effective treatment for shoots and roots formation. For indirect organogenesis, all explant types (leaves, petioles, roots) showed a 100% callusing rate after 2 months in all media variants tested; ODK1: 20 μM thidiazuron (TDZ) + 5 μM indole-3-butyric acid (IBA) or ODK2: 0.5 μM kinetin + 5 μM 2,4-dichlorophenoxy acetic acid (2,4-D). The leaves and petiole explants assured a low rate of shoot regeneration (20%) in ODK1. Afterwards, leaf-, petiole-and root-callus derived from both media were transferred to four new media plant growth regulators—free or with BA + IBA + gibberellic acid (GA_3_). After 10 months from callus transferring, the petiole callus gave rise to roots (20–75%) while the leaf callus exhibited 10–30% shoot or 30% root regeneration. In this study, indirect organogenesis of *O. dictamnus* was carried out for the first time, thus various organs can be used for plant regeneration, and the developed protocol may be applicable in the horticulture industry.

## 1. Introduction

*Origanum dictamnus* L. (Lamiaceae) or Cretan dittany is a native and range-restricted Greek chamaephyte endemic to the island of Crete (Iraklio, Mt Psiloreitis) that grows in habitats such as cliffs, rocks, walls, ravines, and boulders [1]. The conservation status of this taxon based on the IUCN category has been declared as Near Threatened, being under protection by the Greek Presidential Decree 67/81 [1]. *O. dictamnus* represents the most promising case of an endemic neglected and underutilized plant due to the very high potential (94.4%) in the medicinal–cosmetic sector coupled with very high feasibility for sustainable exploitation (91.67%), outlining the extant value chain and the sustainable commercial exploitation associated with this taxon, as already achieved mostly in Crete but also abroad [2]. Finalized monographs and the European Medicines Agency have contented and endorsed the medicinal properties and recommendations of this taxon [3]. Dittany infusion was a complex mixture consisting of several flavones, flavonols, and hydroxycinnamic acid derivatives, partially related to its antioxidant/antiradical activity, being very active as an antiglycative agent, enhancing the functional properties of the Cretan tea beverage, to be used as a food supplement useful in chronic and degenerative disorders [4].

Many species of the genus *Origanum* growing as wild populations in the Mediterranean basin have frequently become the object of predatory exploitation because of overharvest practices seriously threatening the sustainability of these resources, therefore cultivation of these species is highly recommended to lessen their overexploitation [5]. This suggests that the plant could be used as an ornamental in the landscaping industry, or used practically in the pharmaceutical or food industry after the development of protocols for its multiplication through traditional or modern biotechnological methods. Conventional propagation methods are known to face difficulties related to low seed germination rates and moderate rooting percentages in the case of cuttings, pointing out the importance of in vitro propagation to be used as an alternative for large scale multiplication of *Origanum* species, allowing their cultivation and further sustainable use of biological diversity [6]. Plant tissue culture can act as a possible alternative, which may allow rapid propagation for commercial purposes [7] and the development of in vitro cell cultures for accelerating agricultural processes in producing valuable secondary metabolites and other useful phytochemicals [8]. However, there have been limitations in tissue culture systems including the availability and quality of plant materials and exact time of year needed for culture initiation, thus, various parts of a plant such as leaves, petioles, and roots could be used as better explant sources with greater potential due to their abundance and easiness of production as compared to shoot tips [9]. Organogenesis is a process of organ formation such as leaves, shoots, or roots facilitating their regeneration potential from cells and tissues, including two types; the direct one in which cultured explants regenerate shoots without the intermediate stage of callus formation and the indirect one where shoot regeneration occurs only after callus formation [10].

Based on the literature, there is no report on the application of tissue cultures for indirect organogenesis of *O. dictamnus* L. In this context, the main aim of this research was to study the direct and indirect organogenesis potential of the Cretan dittany. In terms of direct organogenesis, the specific objectives were (1) to identify an improved medium, thus two basal culture media were included: Murashige and Skoog (MS) [11], and Gamborg B5 (B5) [12] and (2) to optimize the existing micropropagation protocol and to maximize the ex vitro survival success of shoot node explants according to a previous study conducted by Sarropoulou et al. [13], herein under the effect of different cytokinin types and concentrations. In the framework of indirect organogenesis, this study was oriented (1) to evaluate callus induction, shoot and/or root regeneration potential of different tissue parts (leaves, petioles, roots) under the effect of different plant growth regulators (PGRs) and (2) to elucidate whether leaf-, petiole- and root-callus explants as induced in the previous stage could be further differentiate into new callus, shoots, and/or roots.

## 2. Results

### 2.1. In Vitro Shoot Proliferation—Rooting and Ex Vitro Acclimatization

Among the 14 treatments tested, the MS medium containing 2.2 μM BA exhibited the highest proliferation rate (3.15) and root number (16.5 roots/rooted explant), while the B5 medium enriched with 2.2 μM BA gave the highest percentage in the formation of new multiple shoots (90%); as reported in the previous work [13]. The MS medium enriched with either 2.2 μM BA or 2.32 μM Kin gave higher shoot lengths (2.66–2.78 cm) and root lengths (1.73–2.07 cm) as compared to the other 12 treatments. Statistically non-significant differences were observed in the number of new shoots/explants among the 14 treatments (Figure 1a–h).

Taking simultaneously into consideration all proliferation and rooting parameters, the MS medium was the ideal one, BA the preferred cytokinin type, and 2.2 μM its optimum concentration, promoting better direct organogenesis of explants in a single 30-day culture stage (3.15 proliferation rate, 85% shoot formation, 2.78 cm shoot length, 100% rooting, 16.5 roots/rooted explant 2.07 cm long; as shown in the previous work [13] without occurrence of necrosis (0%) (Figure 2a,b). A survival rate of 100% was achieved in the case of plants derived in vitro from the B5 medium enriched with 2.46 μM 2-iP. Additionally, survival rates of 80–100% were recorded for rooted plants derived in vitro from B5 medium and 66.67–86.67% for those derived from the MS one, exhibiting a final 80.4% ex vitro survival rate (mean value) after 2 months in the greenhouse mist regardless of previous in vitro treatment (medium type, cytokinin type, cytokinin concentration) (Figure 2c,d).

### 2.2. In Vitro Culture of Different Plant Tissue Explants (Leaves, Petioles, Roots)

After 2 months of culture, callus formation was noticed in all treatments. Regeneration of shoots at a rate of 20% on leaf and petiole callus in ODK1 was observed, while no shoot regeneration occurred in the other treatments. Leaf, petiole and root explants cultured in ODK2 medium did not result in shoot regeneration. No treatment led to callus, shoot and root induction at the same time regardless of initial explant type (Figure 3a–n, Table 1).

### 2.3. In Vitro Culture of Different Callus Explants (Leaf-, Petiole-, Root-Callus)

In the following stage, after 10 months of culture, formation of new secondary callus at a 30% rate on initial primary root callus was observed after transferring the explants from ODK2 (0.5 μM Kin + 5 μM 2,4-D) to ODR0 (PGR’s-free) medium while petiole callus exhibited 60% and 75% new callus formation rates when transferred from ODK2 to ODR3 (2.2 μM BA + 0.25 μM IBA + 0.3 μM GA_3_) and ODR0 media, accordingly. Shoot regeneration from leaf callus was evident at a rate of 10% and 30% when leaf callus transferred from ODK1 (20 μM TDZ + 5 μM IBA) to ODR0 and ODR2 (1.1 μM BA + 0.25 μM IBA + 0.3 μM GA_3_) medium, respectively. Root regeneration from leaf callus (30%) was recorded when explants derived from ODK2 and cultured afterwards in ODR0. Petiole callus originated from ODK2 were afterwards differentiated and gave regeneration to roots at a 20–75% rate in ODR0 (75%), ODR1: 0.25 μM IBA + 0.3 μM GA_3_ (20%), ODR2 (20%) and ODR3 (70%) media. Among the three callus explant types, petiole callus had the best root regeneration performance followed by leaf callus. Root callus did not have any shoot and/or root organogenetic response, while leaf callus led to shoot or root regeneration when derived from ODK1 or ODK2, respectively (Figure 4a–g, Table 2).

## 3. Discussion

### 3.1. Micropropagation and Ex Vitro Acclimatization

Shoot proliferation and rooting in vitro is highly contingent on the composition of the basal medium in salts including type, concentration and macronutrients/micronutrients ratio, source of iron, vitamins and surrounding conditions such as photoperiod and temperature [14], therefore different media used in this research with Cretan dittany (MS, B5) have different composition and concentration of constituents, leading to different responses towards plant growth. The augmented potential requirement of *O. dictamnus* shoot node explants for proliferation, rooting and subsequent growth could explain its better performance in the MS medium than in B5. The different responses of explants to the two basal culture media studied herein could be ascribed to the different NO_3_^−^/NH_4_^+^ ratio in each medium or to the difference in total nitrogen content, which is 39.4 mM (18.79 mM KNO_3_ + 20.61 mM NH_4_NO_3_) in MS, and 25.74 mM in B5 (24.73 mM KNO_3_ + 1.01 mM NH_4_2SO_4_) [15]. The MS proved to be the ideal basal medium enhancing shoot proliferation in other *Origanum* species as well including *O. onites* [16], *O. syriacum* L. [17], *O. majorana* L. [18], *Origanum vulgare* x *applii* [19], and *O. vulgare* [20].

Except the basal culture medium composition, the setting up of a competent in vitro direct regeneration protocol is also strongly dependent on the quest of the best cytokinin type and its best concentration as a crucial success coefficient whichever the plant species is [21]. Shoot proliferation performance of *O. dictamnus* microshoots under study was better enhanced by 2.2 μM BA followed by 2.32 μM kinetin and afterwards by 2.46 μM 2-iP. The height of the in vitro grown shoots, the increase in the length of the internodes, the rate of the proliferated microshoots, and above all, the genotype are parameters which are highly affected by the individual and combined application of the type and concentration of the added cytokinin to the culture medium [22]. Differences that are due to stability, mobility, conjugation and hormone oxidation rates among the different cytokinins could explain the disparate competence of explants to different cytokinin-relative strengths during the reinforcement of the shoot induction stage [23]. The supremacy of BA in relation to kinetin and 2-iP in prompting multiple shoot production has been demonstrated, presumably firstly because the chemical stability of BA is the highest among the group of cytokinins in in vitro plant production systems whereas the majority of other cytokinins, especially of the purine-type are deemed of lower stability [24]. Secondly, the quantity of BA declined in the medium is of lower rate as compared to the other phytohormones or is not facile easily disintegrated and therefore carries on in the medium, allowing greater amounts of free or ionized BA be promptly disposable to plant tissues through the medium [25]. Thirdly, there is the higher discontinuating effect of BA on lifting the dominance of the shoot apex and switching on the lateral shoots configuration and shoot tip cultures leaf vegetative evolvement [23]. Finally, plant tissues have a higher ability to metabolize BA more readily than other synthetic growth regulators [26]. Numerous studies support the higher effectiveness of BA, as the most appropriate cytokinin type applied alone or in combination with an auxin (IBA, NAA) for in vitro proliferation in several other *Origanum* species, including *O. majorana* L. [18], *O. sipyleum* [27], *O. syriacum* and *O. ehrenbergii* [6], *O. vulgare* [28], *O. minutiflorum* [29], *O. vulgare* x *applii* [19] and *O. vulgare* [20].

Rooting performance of *O. dictamnus* microshoots under study was better stimulated by the cytokinin BA in the MS medium followed by kinetin and subsequently by 2-iP, while in B5 medium kinetin was the most preferred cytokinin, BA the least preferred and 2-iP of moderate ability, each cytokinin applied with 0.25 μM IBA + 0.3 μM GA_3_. Different responses have been outlined in other *Origanum* species where rooting of microshoots was better promoted either in a full-strength MS medium PGRs-free such as in the case of *O. syriacum* and *O. ehrenbergii* [6] or in different MS or B5 strength media supplemented only with auxins (IBA, NAA, IAA) in the absence of cytokinins including *O. sipyleum* L. [27,30] and *O. acutidens* (Hand.-Mazz.) Ietswaart [7], *O. syriacum* [17,31], *O. heracleoticum* L. [32] and *O. vulgare* [33]. The percentage of rooted microshoots is a coefficient of greater importance in comparison to root elongation for plantlets’ settlement and effectual hardening outcome to ex vitro conditions, nevertheless more lengthy roots permit better fastening of plantlets in the substrate and more efficient exploitation of water potential and minerals due to greater easiness and higher rate of absorption [34]. Rooted Cretan dittany plantlets grown on a peat: perlite (1:1 *v*/*v*) substrate mixture after 2 months in the greenhouse mist gave a mean 80.4% ex vitro survival rate, regardless of previous in vitro conditions. A wide range of survival rates (48–98%) of rooted in vitro plantlets to ex vitro conditions during acclimatization and gradual hardening process is reported for other *Origanum* species including *O. sipyleum* L. [30], *O. majorana* L. [18], and *O. syriacum* L. [17] as well as a wide range of substrate mixtures such as peat: perlite (1:1, 1:2, 1:3, 2:1 *v*/*v*) [6,30,31,32]; peat moss: vermiculite (1:1: *v*/*v*) [18]; peat moss: sand: vermiculite (1:1:1 *v*/*v*) [17]; and peat: sand: perlite (1:1:1 *v*/*v*) [30]. Therefore, between the two basal media (MS, B5), the three cytokinin types (BA, kinetin, and 2-iP) and the three cytokinin concentrations tested, the MS proved to be the ideal medium, BA the most effective cytokinin and 2.2 μM the optimum concentration combined with 0.25 μM IBA and 0.3 μM GA_3_ promoting the best proliferation and rooting in vitro of shoot-node explants of *O. dictamnus* in a single 30-day culture stage ensuring 100% ex vitro survival success. The high ex vitro acclimatization and hardening survival rate of in vitro plantlets and their following vegetative and root system growth is of paramount importance in every plant tissue culture system since it can contribute to cost-savings, marketing trade cycle abridge, speeding up the installation of propagation material, and the further introduction of the plants into cultivation in field conditions [35].

In a previous study on the same plant species [13] as presented herein, among the two different basal media (MS and B5) and the two different BA concentrations (1.11 and 2.22 μM) applied simultaneously with 0.25 μM IBA and 0.3 μM GA_3_, the MS medium + 2.22 μM BA found to be superior for shoot proliferation and rooting in vitro as well as for ex vitro survival success of the rooted microshoots. The further advancements of this study were the effect of two other cytokinins types except of BA, that of kinetin and 2-iP and the fact that all these three cytokinins were supplemented in MS and B5 media at two different and marginal concentrations, the lower of 0 μM, (cytokinin-free) and the higher of 4.4 μM, which had not been tested in the previous work of Sarropoulou et al. [13]. The findings of this study illustrate that the increase in all shoot proliferation and rooting in vitro parameters, except of shoot number in MS medium under the 2.2 μM BA + 0.25 μM IBA + 0.3 μM GA_3_ combination, as well as in the ex vitro survival rate, can be the result of an optimal division and elongation reinforcement due to the role of BA as a cytokinin mainly in cell division/proliferation and less to cell expansion, while IBA as an auxin in both cell division and cell enlargement [21], compared to the other two BA concentration combinations, 0 and 4.4 μM. In parallel with our findings, in another *Origanum* species, in *O. acutidens* (Hand.-Mazz.) Ietswaart, each increase in BA concentration (0.6–2.4 mg/L) with 0.2 mg/L NAA led consistently to increased root numbers, shoot and root lengths up to 1.8 mg/L BA, however a sharp decline in all the above-mentioned parameters was observed when higher BA concentration (2.4 mg/L) was used [7] and this response could be attributed to the declined absorption rate of water and mineral components from the medium [36]. It becomes evident that 2.2 μM herein is the optimum BA concentration for a cytokinin and endogenous auxin balance leading to release of the shoot apical dominance [37], thus besides the cytokinin/auxin balance there is a strong inter-correlation between the endogenous concentrations of PGRs and the applied exogenous ones in the medium affecting morphogenesis [38]. As concerns rooting, the effect of BA on root growth was concentration-dependent [39] since root meristems were enlarged and gave rise to faster growing and more branched roots under treatment with cytokinins, especially BA which regulates the meristem activity due to the increased expression of cytokinin oxidase [40]. The augmented mitotic activity in plants BA-induced has been associated with higher radicle length and higher number of initiated roots [41].

### 3.2. In Vitro Culture of Different Plant Tissue Explants (Leaves, Petioles, Roots)

The process of plant organs regeneration is controlled by the appropriate concentration ratio between auxins and cytokinins, the two main PGRs groups that are essential to drive specific organogenesis responses from different tissue parts (leaf, petiole, root, etc.), thus the interplay actions between cytokinins and auxins are complex and roots, shoots and callus formation are highly affected by different combinations and concentrations of PGRs [42], explant type and plant species [43]. The results of this study showed 100% callus induction in all cases, irrespective of explant type, PGRs types, and concentrations. An explanation for the callusing response herein could be the increase of the endogenous auxin level on the cutting edge of each tissue part (leaf, petiole, root), which switches on cell proliferation, especially in media fortified with cytokinins [44]. In several *Origanum* spp. (*O. vulgare*, *O. vulgare* subsp. *hirtum*, *O. syriacum*) [45], the MS medium supplemented with PGRs proved to be effective for callus production from either leaf explants [45,46] or from hypocotyl explants in marjoram (*O. majorana* L.) [47].

The medium composition and explant source as a combined factor exerted a significant effect on callus initiation and plant regeneration [48]. In the studied *O. dictamnus*, a simultaneous occurrence of callus formation (100%) and shoot regeneration (20%) was obtained by leaf and petiole explants in ODK1 (20 μM TDZ + 5 μM IBA) medium. TDZ is a phenylurea that impels a high morphogenesis activity in plant tissues especially in leaves [49] and actively participates in auxin synthesis by raising the endogenous IAA level and its precursor tryptophan [50], displaying different regeneration responses dependent on genotype and initial explant type [49]. In tissue culture systems, TDZ alone or combined with IBA has been found to be effective for shoot regeneration responses [49]. TDZ has been shown to exert both cytokinin and auxin like activity as compared to other commonly used auxins and cytokinins, owing to its ability to shift the level of endogenous PGRs [51] and this ability might be due to the speed up that causes in endogenous cytokinin production rate and inhibition in the activity of cytokinin oxidase enzymes [52], ensuring better nutrients absorption and increased regeneration [53].

### 3.3. In Vitro Culture of Different Callus Explants (Leaf-, Petiole-, Root-Callus)

The regeneration ability of callus is dependent on plant species, explant type, medium components and endogenous concentrations of PGRs as an outcome of their uptake from extracellular sources, and their metabolism and endogenous interaction [54]. Even though plant cells convey the same genetic make-up information, their morphogenetic competence is not consistent based on differences on the one hand to the spatial and temporal distribution of the cells and on the other hand to their physiological and developmental stage [55]. In the studied *O. dictamnus*, after 10 months of culture, there was new secondary callus formation over the initial primary ones only by root callus when transferred from ODK2 (0.5 μM kinetin + 5 μM 2,4-D) to ODR0 medium (PGR’s-free), and by petiole callus from ODK2 to ODR3 (2.2 μM BA + 0.25 μM IBA + 0.3 μM GA_3_) or ODR0.

Previous studies have demonstrated that the cytokinin BA at a certain concentration can boost callus subculture, differentiation, and regeneration in a large number of plant species [56]. In this study, shoot regeneration occurred only through leaf derived callus when transferred from ODK1 to either ODR0 medium (PGRs-free) or ODR2 (1.1 μM BA + IBA + GA_3_) medium. The low shoot regeneration rate from leaf callus (10–30%) might be due to the long-term maintenance period (10 months) of the explants in culture [57]. In vitro shoot formation may be subjected to change depending upon the explant types used [58]. Ehsandar et al. [59] pointed out the differentiation ability of calli into shoot primordia after repeated subcultures in MS medium containing BA. Leaf explants sourced calli demonstrated shoot regeneration ability versus petiole and root callus explants of *O. dictamnus* as presented herein and the reasons for this response might be (1) the higher juvenility and the lower number of vascular tissue in leaf-callus explants [60], (2) young leaves are healthy, non-senescing, nutrient-rich tissues that contain higher concentrations of endogenous hormones [61] and (3) leaf cells have higher organogenic potential than petiole and root cells [62]. Root regeneration from *O. dictamnus* leaf-callus derived explants was observed in ODK2-ODR0 (PGRs-free) transition combination media whereas petiole-callus explants differentiated into roots in all tested regeneration media, both in the absence of PGR’s and enriched with BA + IBA + GA_3_. The in vitro shoot regeneration of *O. dictamnus* through leaf-derived callus and petiole-derived callus and not through callus of root origin is a very important step because the vegetative parts of a plant are desirable explants for tissue culture systems due to their ability to preserve the genetic homozygosity of the parent genotype [63].

In this study with *O. dictamnus*, root callus explants in the ODR0 (PGRs-free) medium after successive subcultures became dark-brown, then black, and finally collapsed. In consistency with our findings, hypocotyl callus of marjoram (*O. majorana* L.) retained high proliferation rate for two subcultures but afterwards the callus grew slower, turned brown, and did not survive in subsequent subcultures [47]. Such retardation may be related to the oxidation of phenolic compounds [64] that inhibit the activity of some essential enzymes, thus suppressing in vitro proliferation [65]. This explanation could also justify the survival failure of *O. dictamnus* desiccated shoots regenerated from leaf-callus in ODK1 and then transferred to ODR0 or ODR2 media.

## 4. Materials and Methods

### 4.1. Plant Material and Culture Conditions

After taxonomic identification of *O. dictamnus* plants conserved ex situ in the premises of the Balkan Botanic Garden of Kroussia, Institute of Plant Breeding and Genetic Resources, Hellenic Agricultural Organization—DIMITRA, seed collection took place from mature inflorescences and these seeds were given thereafter the unique IPEN (International Plant Exchange Network) accession number GR-1-BBGK-03,2108. All information regarding the disinfection protocol (fungicide, ethanol, NaOCl) followed, the initial establishment conditions of 7-year-old seeds after long-term storage (4 °C, RH < 5%) and 37.93% maximum germination rate obtained under 16 h photoperiod and 21–23 °C temperature after 12 days of in vitro culture in MS medium enriched with 20 g L^−1^ sucrose and 6 g L^−1^ Plant Agar are elaborately described in a previous study [13]. The sub-culturing and transferring of the in vitro stock cultures to fresh nutrient medium were taking place every 4 weeks. In all experiments, culture media were sterilized through autoclave at a temperature of 121 °C for 20 min after adjustment of the pH value of the media to 5.8 using dilute solutions of 1 N KOH and 0.02% HCL in drops for balancing until reaching the exact value. Magenta vessels (Baby food jars, autoclavable, reusable, 62.4 mm × 95.8 mm, 200 mL, Sigma-Aldrich, Merck KGaA, Germany) sealed with Magenta^TM^ B-caps were used for the placement of the nutrient medium and the culture of the explants. Each Magenta vessel was filled with 25 mL of medium. A growth chamber under controlled laboratory conditions of temperature 22 ± 2 °C, photoperiod 16 h light/8 h dark, illumination type of cool white fluorescent lamps (WFLs) (PHILIPS, 36 W/830 G13 1214 mm) and 40 μmol m^−2^ s^−1^ light quantity was used for the incubation of the in vitro plant tissue cultures.

### 4.2. In Vitro Direct Organogenesis and Ex Vitro Acclimatization

For direct organogenesis, internodal shoot segments 1–1.5 cm long that bore two buds were the experimental explants stem from already established in vitro cultures. The concurrent application of two basal nutrient media (MS, B5) with three cytokinin types, i.e., BA, Kin and 2-iP was evaluated. Each cytokinin was incorporated into the basal culture medium, either MS or B5 at three concentrations, i.e., BA: 0, 2.2 and 4.4 μM, Kin: 0, 2.32 and 4.65 μM, and 2-iP: 0, 2.46 and 4.92 μΜ. All media were filled with 0.25 μM indole-3-butyric acid (IBA) as an auxin type boosting root formation, 0.3 μM gibberellic acid (GA_3_) for shoot elongation, and 20 g L^−1^ sucrose (Duchefa Biochemie, The Netherlands) as the main energy carbon source and afterwards were gelled with Plant Agar (Duchefa Biochemie, The Netherlands) at 6 g L^−1^. The basic environmental requirements for incubation of the in vitro plant cultures inside the controlled laboratory growth chamber were set as follows: 22 ± 2 °C temperature, 16 h photoperiod, WFLs of 40 μmol m^−2^ s^−1^ light irradiance. The experiment was terminated after 30 days, wherein eight different macroscopic attributes for shoot proliferation and rooting potential were evaluated.

In early January, microshoots with well-developed root systems after washing with tap water to remove any adhering solidifying agent were then transferred in the internal heated mist of the greenhouse within multiple propagation trays each of 100 mL volume composed of peat (Terrahum, Klasmann) and perlite (Geoflor) substrate mixture at a 1:1 *v*/*v* ratio. The prevailing environmental conditions (temperature—T, relative humidity—RH%, decreased light intensity) inside the greenhouse mist system were 18 °C base T for plantlets root system, 15 °C air T for the vegetative part of the plants and RH ranging between 80% and 100% under conditions of thermal curtains. After 60 days period time in the mist for gradual acclimatization, on early March, the survival rate of the successfully undergo hardening plants was recorded. The following step included the transplantation of the plants into bigger pots of 0.33 L volume capacity (dimensions: 8 cm length × 8 cm width × 7 cm height) supplemented with peat moss (TS2, Klasmann) and perlite mixture substrate at 3:1 *v*/*v*, respectively. The newly transplanted plants were then transferred outside the indoor mist system in the bench of the same greenhouse without heating where the prevailing environmental conditions were scheduled to be 17–24 °C temperature and 55–70% RH range providing a progressively decrement of RH about 5% per day and a gradual increment of light intensity for a period of 3 months for further growth. In the subsequent stage during the onset of summer, particularly in early June due to increased temperatures inside the greenhouse, the vegetative and root system developed plants were transplanted into 2.5 L higher volume pots filled with peat (TS2), perlite and soil (2:1:½*v*/*v*) and placed in the natural environment, in the outdoor nursery under 50% direct sun exposure provided by a shading net for ex situ conservation purposes and future sustainable exploitation.

### 4.3. In Vitro Culture of Different Plant Tissue Explants (Leaves, Petioles, Roots)—Indirect Organogenesis

Three different explant types were used: (1) leaves, (2) petioles, and (3) roots, cultured in MS medium with two different PGRs type and concentration combinations; ODK1: 5 μM IBA + 20 μM thidiazuron (TDZ), and ODK2: 5 μM 2,4-dichlorophenoxy acetic acid (2,4-D) + 0.5 μM Kin. These two media (ODK1 and ODK2) were also enriched with 30 g L^−1^ sucrose and 6 g L^−1^ Plant Agar. After two months, callus, shoot and root regeneration rates (%) were recorded.

In the following stage, the callus derived from the three explant types (leaves, petioles, roots) in ODK1 and ODK2 media were then transferred in four new PGRs-supplemented MS media for further regeneration (ODR0, ODR1, ODR2, and ODR3), all enriched with 20 g L^−1^ sucrose and 6 g L^−1^ Plant Agar. The composition of the four new regeneration media was: ODR0 (PGR’s-free), ODR1: 0.25 μM IBA + 0.3 μM GA_3_, ODR2: 1.1 μM BA + 0.25 μM IBA + 0.3 μM GA_3_, and ODR3: 2.2 μM BA + 0.25 μM IBA + 0.3 μM GA_3_. Aggregates of brittle and very soft texture sponge-like callus were used as experimental explants, cultured under 24 h darkness at 21–23 ^o^C. After 10 months, the following data were recorded: new callus formation percentage (%): the number of initial callus explants with formation of new callus over initial ones/the total number of initial callus explants × 100%, shoot regeneration percentage (%): the number of callus explants with multiple shoot induction/the total number of callus explants × 100%, and root regeneration percentage (%): the number of callus explants with root formation/the total number of callus explants × 100%. Detailed chemical compositions of the different culture media tested for indirect in vitro organogenesis subsequent stages are presented in Table 3.

### 4.4. Statistical Analysis

All experiments were completely randomized, and data were analyzed with the use of the SPSS version 17.0 statistical package (SPSS Inc., Illinois, New York, USA). In Tables and figure graphs, the analysis of variance (one-way ANOVA) and the Duncan multiple-range test at a 5% level were deployed for mean values ± standard errors (S.E.) variability and comparison among treatments for each parameter per experiment conducted. In tables, means ± S.E. with different letters within a column are statistically significant different from each other according to the Duncan’s multiple range test at *p* ≤ 0.05 (Table 1 and Table 2). In figures representing results data in the form of graphs, different bars per graph denoted by different letters indicate statistically significant differences among treatments (*p* ≤ 0.05) and error bars in each bar per graph are standard errors. For each figure graph, a horizontal scale bar on the *X*-axis is provided in its respective caption in parenthesis (Figure 1a–h and Figure 2c). Furthermore, in figures representing photo images and not graphs, a scale bar of 1 cm is provided on each multi-plate image (Figure 2a,b,d, Figure 3a–h and Figure 4a–g).

The micropropagation experiment included 14 treatments with 12 repetitions/treatment (four explants/vessel × three vessels/treatment), therefore it was a 2 × 3 × 3 factorial one with two culture media (MS, B5), three cytokinin types (BA, kin, 2-iP) and three cytokinin concentrations.

The first stage of the indirect organogenesis experiment (six treatments) was a 3 × 2 factorial one including three explant types (leaves, petioles, roots) and two culture media (ODK1, ODK2) with 25 explants per treatment and per explant type (5 groups × 5 explants/vessel, i.e., 25 leaves, 25 petioles and 25 roots).

The second stage of indirect organogenesis experiment was a 3 × 4 × 2 factorial one consisting of 24 treatments corresponding to three callus explant types (leaf-callus, petiole-callus and root-callus), four new tested culture media (ODR0, ODR1, ODR2, ODR3), and two media initially used (ODK1, ODK2) prior to callus formation, i.e., 3 groups (vessels) × 10 calluses per vessel, thus 30 leaf calluses, 30 petiole calluses and 30 root calluses per treatment.

## 5. Conclusions

An efficient in vitro indirect regeneration protocol of the Cretan dittany was carried out and is reported for the first time in the present study. The unravelling and accomplishments of direct and indirect regeneration potential of various plant tissues and organs of *O. dictamnus* are deployed and underlined. In particular, from the results, it was observed that MS medium provides a better response for micropropagation and ex vitro survival of *O. dictamnus* when compared with Gamborg B5. It is clearly illustrated that between different cytokinin types and concentrations tested, there is a strong inter-correlation and interaction. Among the 14 combination treatments tested for direct organogenesis, the 2.2 μM BA + 0.25 μM IBA + 0.3 μM GA_3_ was the most cost- and time-effective one within a single-one stage (30 days). Among the explant types and the culture media tested, shoot regeneration occurred only by leaf- and petiole explants cultured in medium containing 20 μM TDZ + 5 μM IBA. Petiole callus had the best root regeneration performance followed by leaf callus, and leaf callus resulted in shoot or root regeneration. *O. dictamnus* callus can be utilized for obtaining active constituents, nevertheless, significant scope exists in establishing cell cultures for enhanced production of the desired metabolite in a controlled environment. Further studies could be carried out on the analysis of secondary metabolites produced in vitro. Selected clones with higher bioactive compounds content could satisfy pharmaceutical industry needs, reducing overexploitation of this range-restricted/threatened Greek endemic from nature for plant material sourcing.

## Figures and Tables

**Figure 1 plants-12-00182-f001:**
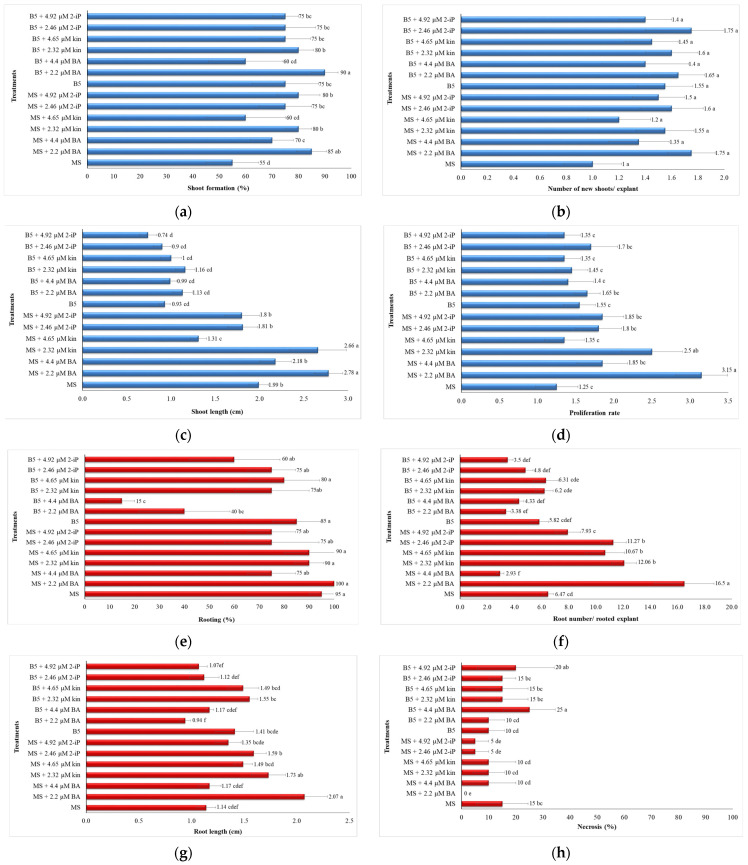
Effect of two basal culture media (MS, B5) supplemented with 0.25 μM IBA, 0.3 μM GA_3_, 20 g L^−1^ sucrose and 6 g L^−1^ Plant Agar (pH: 5.8) in combination with three cytokinin types (BA, Kin, 2-iP) at three concentrations on shoot proliferation and rooting parameters in *O. dictamnus* explants after 30 days of in vitro culture: (**a**) Shoot formation (%) (graph horizontal scale bar: 1 unit = 10%); (**b**) Number of new shoots/explant (graph horizontal scale bar: 1 unit = 0.2); (**c**) Shoot length (cm) (graph horizontal scale bar: 1 unit = 0.5 cm); (**d**) Proliferation rate (graph horizontal scale bar: 1 unit = 0.5); (**e**) Rooting (%) (graph horizontal scale bar: 1 unit = 10%); (**f**) Root number/rooted explant (graph horizontal scale bar: 1 unit = 2); (**g**) Root length (cm) (graph horizontal scale bar: 1 unit = 0.5 cm); (**h**) Necrosis (%) (graph horizontal scale bar: 1 unit = 10%).

**Figure 2 plants-12-00182-f002:**
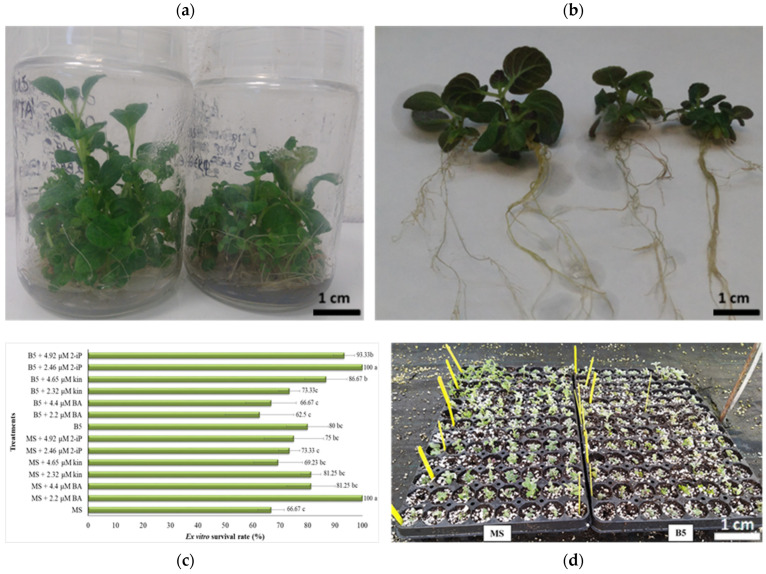
Micropropagation and ex vitro acclimatization of in vitro rooted plantlets of *O. dictamnus* explants: (**a**,**b**) Shoot proliferation and rooting after 30 days of in vitro culture in MS basal nutrient medium supplemented with 2.2 μM BA, 0.25 μM IBA, 0.3 μM GA_3_, 20 g L^−1^ sucrose and 6 g L^−1^ Plant Agar (pH: 5.8) inside and outside vessels, respectively (scale bar: 1 cm); (**c**) Effect of two culture media (MS, B5) supplemented with 0.25 μM IBA, 0.3 μM GA_3_, 20 g L^−1^ sucrose and 6 g L^−1^ Plant Agar (pH: 5.8) in combination with three cytokinin types (BA, Kin, 2-iP) at three concentrations on survival rate (%) (graph horizontal scale bar: 1 unit = 10%); (**d**) Acclimatized plants in a peat: perlite (1:1) substrate in the greenhouse mist (scale bar: 1 cm).

**Figure 3 plants-12-00182-f003:**
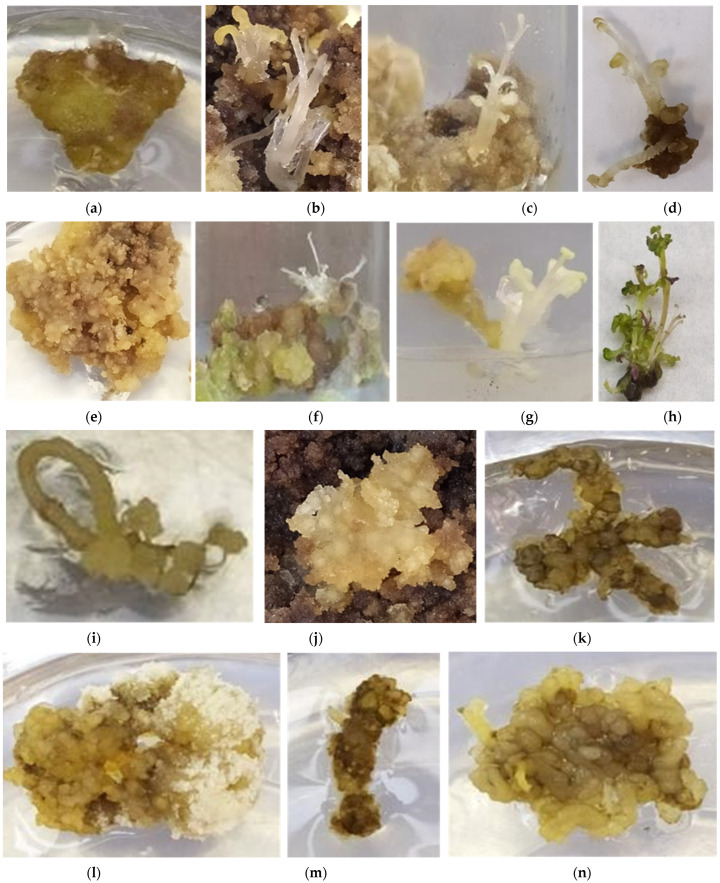
In vitro culture of three explant types (leaves, petioles, roots) of *O. dictamnus* L. in different media after two months: (**a**–**d**) Root, callus and shoot formation from leaf explants in ODK1; (**e**–**h**) Callus and shoot regeneration from petioles in ODK1; (**i**,**j**) Callus formation on root explants in ODK1; (**k**) Callus formation in root explants in ODK2; (**l**–**n**) Callus formation on leaf, petiole and root explants, respectively in ODK2. (Scale bar: 1 cm).

**Figure 4 plants-12-00182-f004:**
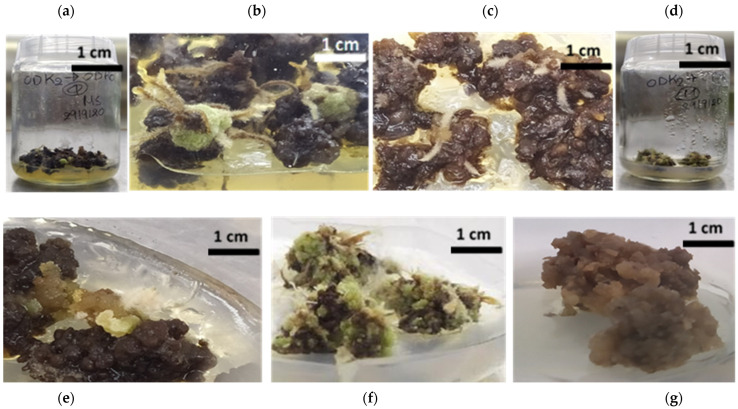
In vitro culture of different callus explant types of *O. dictamnus* in different PGRs-supplemented MS media: (**a**–**c**) New callus and root regeneration from leaf-callus when transferred from ODK2 to ODR0; (**d**–**f**) Differentiation of petiole callus into new callus and roots after transferred from ODK2 to ODR3; (**g**) Root callus turned brown after successive subcultures in ODR0. (Scale bar: 1 cm).

**Table 1 plants-12-00182-t001:** Effect of three different explant types (leaves, petioles, roots) of *O. dictamnus* L. and two different media on callus formation, shoot and root regeneration (%) after a 2-month period.

Treatments						Response (after 2 Months)		
Explant Type	Culture Medium	TDZ (μM)	Kin (μM)	IBA (μM)	2,4-D (μM)	Callus Formation (%)	Shoot Regeneration (%)	Root Regeneration (%)
Leaves	ODK1	20	-	5	-	100	20.0 ± 6.3 a	0
	ODK2	-	0.5	-	5	100	0.0 ± 0.0 b	0
Petioles	ODK1	20	-	5	-	100	20.0 ± 11.0 a	0
	ODK2	-	0.5	-	5	100	0.0 ± 0.0 b	0
Roots	ODK1	20	-	5	-	100	0.0 ± 0.0 b	0
	ODK2	-	0.5	-	5	100	0.0 ± 0.0 b	0

Means ± standard error (S.E.) with different letters within the “Shoot Regeneration (%)” column are statistically significant different from each other according to the Duncan’s multiple range test at *p* ≤ 0.05.

**Table 2 plants-12-00182-t002:** Effect of explant type (leaf callus, petiole callus, root callus) of *O. dictamnus* L. and different media (ODR0, ODR1, ODR2, ODR3) on callus, shoot and root regeneration (%) (10 months).

Medium Prior Callusing	Explant Type	Culture Medium Code Number	BA (μM)	IBA (μM)	GA_3_ (μM)	New Callus Formation (%)	Shoot Regeneration (%)	Root Regeneration (%)
ODK1 (leaves, petioles, roots)	Leaf callus	ODR0	0	0	0	0.0 ± 0.0 d	10.0 ± 5.8 b	0.0 ± 0.0 d
		ODR1	0	0.25	0.3	0.0 ± 0.0 d	0.0 ± 0.0 c	0.0 ± 0.0 d
		ODR2	1.11	0.25	0.3	0.0 ± 0.0 d	30.0 ± 11.5 a	0.0 ± 0.0 d
		ODR3	2.22	0.25	0.3	0.0 ± 0.0 d	0.0 ± 0.0 c	0.0 ± 0.0 d
	Petiole callus	ODR0	0	0	0	0.0 ± 0.0 d	0.0 ± 0.0 c	0.0 ± 0.0 d
		ODR1	0	0.25	0.3	0.0 ± 0.0 d	0.0 ± 0.0 c	0.0 ± 0.0 d
		ODR2	1.11	0.25	0.3	0.0 ± 0.0 d	0.0 ± 0.0 c	0.0 ± 0.0 d
		ODR3	2.22	0.25	0.3	0.0 ± 0.0 d	0.0 ± 0.0 c	0.0 ± 0.0 d
	Root callus	ODR0	0	0	0	0.0 ± 0.0 d	0.0 ± 0.0 c	0.0 ± 0.0 d
		ODR1	0	0.25	0.3	0.0 ± 0.0 d	0.0 ± 0.0 c	0.0 ± 0.0 d
		ODR2	1.11	0.25	0.3	0.0 ± 0.0 d	0.0 ± 0.0 c	0.0 ± 0.0 d
		ODR3	2.22	0.25	0.3	0.0 ± 0.0 d	0.0 ± 0.0 c	0.0 ± 0.0 d
ODK2 (leaves, petioles, roots)	Leaf callus	ODR0	0	0	0	0.0 ± 0.0 d	0.0 ± 0.0 c	30.0 ± 10.0 b
		ODR1	0	0.25	0.3	0.0 ± 0.0 d	0.0 ± 0.0 c	0.0 ± 0.0 d
		ODR2	1.11	0.25	0.3	0.0 ± 0.0 d	0.0 ± 0.0 c	0.0 ± 0.0 d
		ODR3	2.22	0.25	0.3	0.0 ± 0.0 d	0.0 ± 0.0 c	0.0 ± 0.0 d
	Petiole callus	ODR0	0	0	0	75.0 ± 14.4 a	0.0 ± 0.0 c	75.0 ± 2.9 a
		ODR1	0	0.25	0.3	0.0 ± 0.0 d	0.0 ± 0.0 c	20.0 ± 5.8 c
		ODR2	1.11	0.25	0.3	0.0 ± 0.0 d	0.0 ± 0.0 c	20.0 ± 5.8 c
		ODR3	2.22	0.25	0.3	60.0 ± 11.5 b	0.0 ± 0.0 c	70.0 ± 5.8 a
	Root callus	ODR0	0	0	0	30.0 ± 5.8 c	0.0 ± 0.0 c	0.0 ± 0.0 d
		ODR1	0	0.25	0.3	0.0 ± 0.0 d	0.0 ± 0.0 c	0.0 ± 0.0 d
		ODR2	1.11	0.25	0.3	0.0 ± 0.0 d	0.0 ± 0.0 c	0.0 ± 0.0 d
		ODR3	2.22	0.25	0.3	0.0 ± 0.0 d	0.0 ± 0.0 c	0.0 ± 0.0 d

**Table 3 plants-12-00182-t003:** Chemical composition of the different PGRs-supplemented MS basal culture media tested in this study for indirect in vitro organogenesis subsequent stages of *O. dictamnus* L.

PGRs ^1^ (μM)	Culture Medium Code Number					
	ODK1	ODK2	ODR0	ODR1	ODR2	ODR3
TDZ ^2^ IBA ^3^ Kin ^4^ 2,4-D ^5^ BA ^6^ GA_3_ ^7^	20 5 - - - -	- - 0.5 5 - -	- - - - - -	- 0.25 - - - 0.3	- 0.25 - - 1.11 0.3	- 0.25 - - 2.22 0.3

^1^ Plant Growth Regulators (PGRs), ^2^ Thidiazuron (TDZ), ^3^ Indole-3-Butyric Acid (IBA), ^4^ Kinetin (Kin), ^5^ 2,4-Dichlorophenoxy Acetic Acid (2,4-D), ^6^ 6-Benzyladenine (BA), ^7^ Gibberellic acid (GA_3_).

## Data Availability

Not applicable.

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
