# Peer review of "Cretan Dittany (Origanum dictamnus L.), a Valuable Local Endemic Plant: In Vitro Regeneration Potential of Different Type of Explants for Conservation and Sustainable Exploitation"

_plants, 2023, doi:10.3390/plants12010182_

Round 1
Reviewer 1 Report
Dear authors take care of formulation because parts of the paper (abstract, introduction, materials and methods, results, and discussion were published in your previous article https://www.notulaebotanicae.ro/index.php/nbha/article/view/12715/9374 Revise it.
Abstract section
Please revise the abstract, there are more than 200 words.
Page 1 line 20-22 “MS media [ODR0: plant growth regulators free, and ODR1, ODR2, ODR3 with BA (0, 1.11 and 2.22 μM respectively) + 0.25 μΜ IBA + 0.3 μM 22 GA3] for further regeneration” is not clear which are the formula for media variants.
Take care at “In this study, optimized in vitro propagation conditions through direct and indirect organogenesis of O. dictamnus have been carried out and described for the first time.” Because in your recent article (2022) https://www.notulaebotanicae.ro/index.php/nbha/article/view/12715/9374 you also describe the protocol for direct regeneration.
Take care that all “ in vitro” should be written in italics. Check the whole manuscript
Page 9 line 8 correct 2-ip
Page 10, line 48 BA, kinetin 2-iP) put a comma or and after kinetin
Page 12 lines 12, 13, and 21 insert degree sign from the Symbol. Check-in whole document
Line 28 ex situ conservation italics
Line 50 table 3 not table 1
At the Material and methods section has described a protocol for seed germination, but in the Results section, there is nothing about this subject. revise this part
write what is the formula to calculate: new callus formation (%) (percentage of initial callus with formation of new callus), shoot regeneration (%) (percentage of initial callus with shoot 48 induction) and root regeneration (%) (percentage of initial callus with root formation).
At “4.3. In vitro culture of different plant tissue explants (leaves, petioles, roots) - indirect organogenesis” data in the table and text are redundant. I suggest making a table with the composition of all media variants used. And an abbreviation list.
At the Conclusion section you wrote: “An efficient in vitro propagation system of the Cretan dittany is described for the first time for this unique neglected-underutilized Near Threatened range-restricted Greek endemic species with multifaceted commercial exploitation potential in the medicinal, cosmetic and food industry sector” but the same meaning is in your article published in Notulae Botanicae https://www.notulaebotanicae.ro/index.php/nbha/article/view/12715/9374
This sentence ” Nutrient medium plays vital role in propagation of plants through tissue culture” is a general conclusion – please remove
It will be interesting comparative analyses of phytochemical constituents and antioxidant activity between plants regenerated through direct and indirect ways and wild type
Author Response
Journal: Plants (ISSN 2223-7747)
Manuscript ID: plants-2106767
Type: Article
Title: Cretan dittany (Origanum dictamnus L.), a valuable local endemic plant: In vitro regeneration potential of different type of explants for conservation and sustainable exploitation
Authors: Virginia Saropoulou, Eleni Maloupa, Katerina Grigoriadou*
Section: Plant Genetic Resources
Special Issue: Plant Genetic Resources In Vitro Culture: Present Situation and Prospects for Propagation, Conservation and Sustainable Use
We appreciate the anonymous reviewers for their time and effort in reviewing our manuscript and we thank them for their valuable comments which helped us to improve our manuscript in many aspects. We have followed their advice and we have edited the submitted manuscript according to the reviewers’ comments.
All changes made by the authors are yellow highlighted in the revised manuscript.
Response to Reviewer #1 comments
|
Yes |
Can be improved |
Must be improved |
Not applicable |
|
|
Does the introduction provide sufficient background and include all relevant references? |
(x) |
( ) |
( ) |
( ) |
|
Are all the cited references relevant to the research? |
(x) |
( ) |
( ) |
( ) |
|
Is the research design appropriate? |
(x) |
( ) |
( ) |
( ) |
|
Are the methods adequately described? |
(x) |
( ) |
( ) |
( ) |
|
Are the results clearly presented? |
(x) |
( ) |
( ) |
( ) |
|
Are the conclusions supported by the results? |
( ) |
( ) |
(x) |
( ) |
Comments and Suggestions for Authors
Reviewer: Dear authors take care of formulation because parts of the paper (abstract, introduction, materials and methods, results, and discussion were published in your previous article https://www.notulaebotanicae.ro/index.php/nbha/article/view/12715/9374. Revise it.
Authors’ response: We revised the respective parts of the main body text throughout the manuscript as kindly requested so as to be disguisable from the already published. See changes in the revised manuscript with yellow highlighted text, in specific, Abstract (page 1, lines 11-13), Introduction (page 1, lines 31-33 & 38-39), Discussion (page 9 – lines 37-39, 49-52 & page 10 – lines 1-3, 10-13), and M&M (page 12, lines 2-8 & 21-26).
Reviewer: Abstract section, Please revise the abstract, there are more than 200 words.
Authors’ response: We followed the suggested advise by shortening the total abstract text count from 275 words (initial submission) to 204 words (revised version) trying to keep its quality without excluding significant quantitative information. See revised pdf manuscript, page 1, lines 11-24.
Reviewer: Abstract section, Page 1 line 20-22 “MS media [ODR0: plant growth regulators free, and ODR1, ODR2, ODR3 with BA (0, 1.11 and 2.22 μM respectively) + 0.25 μΜ IBA + 0.3 μM GA3] for further regeneration” is not clear which are the formula for media variants.
Authors’ response: In the revised manuscript, in Materials and Methods section, we deleted the initial inserted Table 3 entitled “Detailed description process – scheme of indirect in vitro organogenesis subsequent stages followed in this study with Origanum dictamnus L.” as it was considered redundant and in its turn we added a new Table (named also as Table 3) entitled “Chemical composition of the different culture media tested for indirect in vitro organogenesis subsequent stages followed in this study with O. dictamnus L.” showing clearly the exact formula for media variants and providing below Table as a footnote the abbreviated form of all plant growth regulators (PGRs) tested (see pages 13-14). For the above mentioned reason and for less wording count in the abstract as proposed, we simplified the whole sentence by omitting detailed concentration combinations of PGRs involved. See revised manuscript, page 1, lines 20-21.
Reviewer: Abstract section, Take care at “In this study, optimized in vitro propagation conditions through direct and indirect organogenesis of O. dictamnus have been carried out and described for the first time.” Because in your recent article (2022) https://www.notulaebotanicae.ro/index.php/nbha/article/view/12715/9374 you also describe the protocol for direct regeneration.
Authors’ response: In the revised manuscript, we took care the specific point as indicated and we revised appropriately the last sentence of the abstract. See page 1, lines 22-24.
Reviewer: Take care that all “in vitro” should be written in italics. Check the whole manuscript
Authors’ response: Checked throughout the revised manuscript, see page 4 – line 6, & page 9 – lines 18-19.
Reviewer: Page 9 line 8 correct 2-ip
Authors’ response: We made the appropriate correction “2-iP” instead of “2-ip”, see page 9, line 30.
Reviewer: Page 10, line 48 BA, kinetin 2-iP) put a comma or and after kinetin
Authors’ response: Done, see page 10, line 17.
Reviewer: Page 12 lines 12, 13, and 21 insert degree sign from the Symbol. Check-in whole document
Authors’ response: We checked the whole document and inserted the degree sign from the Symbol menu. See page 12, lines 4, 5, 41, 42 & 50.
Reviewer: Page 12, Line 28 ex situ conservation italics
Authors’ response: Done, see page 13, line 4.
Reviewer: Line 50 table 3 not table 1
Authors’ response: Ok, we corrected the specific point, see page 13, line 28.
Reviewer: At the Material and methods section has described a protocol for seed germination, but in the Results section, there is nothing about this subject. Revise this part.
Authors’ response: We did not provide any text to the results section regarding seed germination in the initial submitted manuscript because it concerns part of already published work by Sarropoulou et al. (2022) [reference list:13], https://www.notulaebotanicae.ro/index.php/nbha/article/view/12715/9374, a citation given from the beginning. However, in the revised manuscript we gave a more detailed description about methodology and main outcome derived based on previous published protocol in M&M section therefore, the whole text was re-written with the relative citation for more clarity so as to be more comprehensible. See page 12, lines 2-8.
Reviewer: Write what is the formula to calculate: new callus formation (%) (percentage of initial callus with formation of new callus), shoot regeneration (%) (percentage of initial callus with shoot induction) and root regeneration (%) (percentage of initial callus with root formation).
Authors’ response: We wrote the calculation formulas for all three percentages terms, see M&M section, page 13, lines 22-28.
Reviewer: At “4.3. In vitro culture of different plant tissue explants (leaves, petioles, roots) - indirect organogenesis” data in the table and text are redundant. I suggest making a table with the composition of all media variants used. And an abbreviation list.
Authors’ response: We took into consideration all the suggestions and we made all the revisions as kindly requested. See pages 13-14.
Reviewer: At the Conclusion section you wrote: “An efficient in vitro propagation system of the Cretan dittany is described for the first time for this unique neglected-underutilized Near Threatened range-restricted Greek endemic species with multifaceted commercial exploitation potential in the medicinal, cosmetic and food industry sector” but the same meaning is in your article published in Notulae Botanicae https://www.notulaebotanicae.ro/index.php/nbha/article/view/12715/9374
Authors’ response: In the revised manuscript, this sentence was deleted and in its turn a new inserted text was given in order to provide different meaning and outline better the progress and advancements obtained. See page 15, lines 2-14.
Reviewer: This sentence “Nutrient medium plays vital role in propagation of plants through tissue culture” is a general conclusion – please remove.
Authors’ response: Done.
Reviewer: It will be interesting comparative analyses of phytochemical constituents and antioxidant activity between plants regenerated through direct and indirect ways and wild type.
Authors’ response: In the revised manuscript, we added in the conclusion section as the last sentence new text which highlights the ongoing research status and further achievements in the same plant species by the authors of this article paper in the production of bioactive compounds, such as the rosmarinic acid using an adventitious root in vitro culture system in bioreactor systems, which is in underway for future publication. See page 15, lines 30-35.

Reviewer 2 Report
Overall, the research has done a lot of work for the first time on tissue culture and regeneration, as well as callus induction and shoot organogenesis of Cretan dittany, and it is recommended acceptance after a major revision.
1. In the abstract, when plant growth regulators (PGRs) first appear, all should be written in full;
2. The abscissa in Figure 1 and Figure 3 marks with the medium and PGR is, they are arranged in slanted, which is neither beautiful nor clear, it is recommended to replace this abscissa with ordinate, and the total coordinate with abscissa;
3. The figures in Figure 2-3 can be combined to form new Figure;
4. Figure 4, all the small figs are placed on the same page;
5. The ABCD of all diagrams is arranged in the diagram, and each diagram should have a bar.
Author Response
Journal: Plants (ISSN 2223-7747)
Manuscript ID: plants-2106767
Type: Article
Title: Cretan dittany (Origanum dictamnus L.), a valuable local endemic plant: In vitro regeneration potential of different type of explants for conservation and sustainable exploitation
Authors: Virginia Saropoulou, Eleni Maloupa, Katerina Grigoriadou*
Section: Plant Genetic Resources
Special Issue: Plant Genetic Resources In Vitro Culture: Present Situation and Prospects for Propagation, Conservation and Sustainable Use
We appreciate the anonymous reviewers for their time and effort in reviewing our manuscript and we thank them for their valuable comments which helped us to improve our manuscript in many aspects. We have followed their advice and we have edited the submitted manuscript according to the reviewers’ comments.
All changes made by the authors are yellow highlighted in the revised manuscript.
Response to Reviewer #2 comments
|
|
Yes |
Can be improved |
Must be improved |
Not applicable |
|
Does the introduction provide sufficient background and include all relevant references? |
(x) |
( ) |
( ) |
( ) |
|
Are all the cited references relevant to the research? |
(x) |
( ) |
( ) |
( ) |
|
Is the research design appropriate? |
(x) |
( ) |
( ) |
( ) |
|
Are the methods adequately described? |
(x) |
( ) |
( ) |
( ) |
|
Are the results clearly presented? |
(x) |
( ) |
( ) |
( ) |
|
Are the conclusions supported by the results? |
(x) |
( ) |
( ) |
( ) |
Comments and Suggestions for Authors
Overall, the research has done a lot of work for the first time on tissue culture and regeneration, as well as callus induction and shoot organogenesis of Cretan dittany, and it is recommended acceptance after a major revision.
- Reviewer: In the abstract, when plant growth regulators (PGRs) first appear, all should be written in full;
Authors’ response: Done, in the revised manuscript, all PGRs firstly appeared in the abstract were written in full followed by their abbreviated form within parentheses. After their full name first mention, the abbreviated form was used. See page 1, lines 11-24.
- Reviewer: The figures in Figure 2-3 can be combined to form new Figure;
Authors’ response: We merged the multi-plate figures in Figure 2-3 (Figure 2a-2b, Figure 3a-3b) of the initial submitted manuscript into a new formed Figure 2 composed of 4 multi-plate figures (Figure 2a, 2b, 2c, 2d) as requested. In addition, changes were made in the caption of the revised Figure 2 by combining and re-arranging text from 2 initial different figure captions, accordingly. See page 4, lines 10-43. Due to this modification, the initial Figure 4 was re-labelled as Figure 3 (see page 4 – line 51 in text & page 7 - line 1 in caption) and so on the initial Figure 5 re-labelled as Figure 4 (see page 7 – lines 22-23 in text & page 8 - line 20 in caption). Thus, the revised manuscript is consisted of 4 Figures instead of 5 Figures during the initial submission.
- Reviewer: The abscissa in Figure 1, and Figure 3a (initial submitted manuscript) but Figure 2c in the revised version marks with the medium and PGR is, they are arranged in slanted, which is neither beautiful nor clear, it is recommended to replace this abscissa with ordinate, and the total coordinate with abscissa;
Authors’ response: We strictly followed the recommendations, thus we made the appropriate re-arrangements i.e. marks with the medium and PGRs not slanted but in a horizontal axis direction, bars instead of columns by replacing in diagrams abscissa with ordinate, and the total coordinate with abscissa in order the final optical picture to be more attractive. In addition, we revised the captions below figures accordingly. See Figures 1a-1g on page 3 (lines 1-44) and Figure 2c on page 4 (lines 25-43) including their captions.
- Reviewer: Figure 4, all the small figs are placed on the same page;
Authors’ response: Done, Figure 4 was re-named in the revised manuscript as Figure 3, so all small multi-plate figs were placed on the same page as suggested, see page 6.
- Reviewer: The ABCD of all diagrams is arranged in the diagram, and each diagram should have a bar.
Authors’ response: Done, see Figure 1a-1g (page 3, lines 1-44, diagram & caption) and Figure 2c (page 4, lines 25-34 in the diagram & 41-43 in the caption below).

Round 2
Reviewer 1 Report
Dear authors,
the abstract section should be revise being too detailed. In my opinion, I would redo it like this: "Origanum dictamnus L. is a medicinal local endemic to the Crete island, Greece. Its propagation through biotechnological tissue culture techniques is essential due to its augmented multi-industrial sector demand. For direct organogenesis, among different culture media variants (MS, Gamborg B5), and cytokinins [6-benzyladenine (BA), kinetin (Kin), 2-isopentenyl adenine (2-iP)], the MS + added with BA (2.2 μM) BA was the most effective treatment for (85% shoot formation, 1.75 shoots 2.78 cm, 3.15and proliferation rate, 100% rooting/survival) and rooting. For indirect organogenesis, all explant types (leaves, petioles, roots) resulted inshowed a 100% callusing rate after 2 months in all media variants tested in ODK1: 20 μM thidiazuron (TDZ) + 5 μM indole 3-butyric acid (IBA) or ODK2: 0.5 μM kinetin + 5 μM 2,4-dichlorophenoxy acetic acid (2,4-D) media. The leaves and petiole explants assure a low rate of Sshoot regeneration (20%) was observed by leaves and petioles in ODK1. Afterwards, leaf-, petiole and root-callus derived from both media were transferred to four new media plant growth regulators-free or with BA + IBA + gibberellic acid (GA3). After 10 months after callus transfering, the petiole callus gave rise to roots (20-75%) while leaf callus exhibited 10-30% shoot or 30% root regeneration. In this study, indirect organogenesis of O. dictamnus was carried out for the first time, thus various organs can be used for plant regeneration, and the developed protocol may be applicable in the horticulture industry."
Page 12 line 7 “are elaborately described in a previously conducted study and recently published by Sarropoulou et al. [13]” delete “conducted” and “recently published by Sarropoulou et al”.
In table 3, you can delete the MS components and carbon source and gelling agent. Let only the plant growth factors.
The conclusion are too long. Please revise it.
The phrase “This modern biotechnological propagation method ensures retrieval of whole, genetically clonal plants from miscellaneous plant tissues and organs (shoot nodes, leaves, petioles, roots, callus) at the same time after 8 approximately 3-4 months through direct and about 12-16 months via indirect organogenesis. The process depicted herein can contribute to diminish genetic erosion and extinction of this Near Threatened plant species in its natural habitat. It is evident that the composition of the basal nutrient culture medium, the type and concentration of the exogenously applied cytokinin, the explant type, and different cytokinin/auxin type and concentration combinations as main and combined factors play a key role on morphogenesis that bring about differential responses related to regeneration potential” is too general. Move in the Discussion section or remove it.
The phrase “A suitable in vitro adventitious root culture system in liquid media at 250 mL Erlenmeyer flasks on a continuous rotary shaker derived from callus cultures initiated from leaf-, petiole- and root-explants in different composition agar-solidified media under different incubation periods for the production of important bioactive molecules, such as rosmarinic acid and further optimization of the methodology for scale up production in a 2 L Balloon Bubble Type Bioreactor (BBTB) system is underway” remove it, is not necessary to be inserted. I said that just because I wish in the future to see this article published.
Author Response
Journal: Plants (ISSN 2223-7747)
Manuscript ID: plants-2106767
Type: Article
Title: Cretan dittany (Origanum dictamnus L.), a valuable local endemic plant: In vitro regeneration potential of different type of explants for conservation and sustainable exploitation
Authors: Virginia Saropoulou, Eleni Maloupa, Katerina Grigoriadou*
Section: Plant Genetic Resources
Special Issue: Plant Genetic Resources In Vitro Culture: Present Situation and Prospects for Propagation, Conservation and Sustainable Use
Response to Reviewer #1 comments
Review Report Form
English language and style
(x) I don't feel qualified to judge about the English language and style
|
Yes |
Can be improved |
Must be improved |
Not applicable |
|
|
Does the introduction provide sufficient background and include all relevant references? |
(x) |
( ) |
( ) |
( ) |
|
Are all the cited references relevant to the research? |
(x) |
( ) |
( ) |
( ) |
|
Is the research design appropriate? |
(x) |
( ) |
( ) |
( ) |
|
Are the methods adequately described? |
(x) |
( ) |
( ) |
( ) |
|
Are the results clearly presented? |
(x) |
( ) |
( ) |
( ) |
|
Are the conclusions supported by the results? |
(x) |
( ) |
( ) |
( ) |
Comments and Suggestions for Authors
Reviewer: The abstract section should be revise being too detailed. In my opinion, I would redo it like this: "Origanum dictamnus L. is a medicinal local endemic to the Crete island, Greece. Its propagation through biotechnological tissue culture techniques is essential due to its augmented multi-industrial sector demand. For direct organogenesis, among different culture media variants (MS, Gamborg B5), and cytokinins [6-benzyladenine (BA), kinetin (Kin), 2-isopentenyl adenine (2-iP)], the MS + added with BA (2.2 μM) BA was the most effective treatment for (85% shoot formation, 1.75 shoots 2.78 cm, 3.15and proliferation rate, 100% rooting/survival) and rooting. For indirect organogenesis, all explant types (leaves, petioles, roots) resulted in showed a 100% callusing rate after 2 months in all media variants tested in ODK1: 20 μM thidiazuron (TDZ) + 5 μM indole 3-butyric acid (IBA) or ODK2: 0.5 μM kinetin + 5 μM 2,4-dichlorophenoxy acetic acid (2,4-D) media. The leaves and petiole explants assure a low rate of Sshoot regeneration (20%) was observed by leaves and petioles in ODK1. Afterwards, leaf-, petiole and root-callus derived from both media were transferred to four new media plant growth regulators-free or with BA + IBA + gibberellic acid (GA3). After 10 months after callus transfering, the petiole callus gave rise to roots (20-75%) while leaf callus exhibited 10-30% shoot or 30% root regeneration. In this study, indirect organogenesis of O. dictamnus was carried out for the first time, thus various organs can be used for plant regeneration, and the developed protocol may be applicable in the horticulture industry."
Authors’ response: We took into consideration the text proposed, thus the abstract was revised accordingly. See 2nd revised manuscript, page 1, lines 11-26.
Reviewer: Page 12 line 7 “are elaborately described in a previously conducted study and recently published by Sarropoulou et al. [13]” delete “conducted” and “recently published by Sarropoulou et al”.
Authors’ response: Done as kindly suggested. See page 10, line 46.
Reviewer: In table 3, you can delete the MS components and carbon source and gelling agent. Let only the plant growth factors.
Authors’ response: We deleted components (macronutrient, micronutrients, vitamins), carbon source (sucrose), gelling agent (Plant Agar) and pH value as proposed. The revised Table 3 includes only the different plant growth regulators per each culture medium tested. In this framework, we modified the caption of the Table, accordingly. See page 12, lines 18-26.
Reviewer: The conclusion are too long. Please revise it.
Authors’ response: We shortened the conclusion section. See revised text in page 13, lines 14-33.
Reviewer: The phrase “This modern biotechnological propagation method ensures retrieval of whole, genetically clonal plants from miscellaneous plant tissues and organs (shoot nodes, leaves, petioles, roots, callus) at the same time after 8 approximately 3-4 months through direct and about 12-16 months via indirect organogenesis. The process depicted herein can contribute to diminish genetic erosion and extinction of this Near Threatened plant species in its natural habitat. It is evident that the composition of the basal nutrient culture medium, the type and concentration of the exogenously applied cytokinin, the explant type, and different cytokinin/auxin type and concentration combinations as main and combined factors play a key role on morphogenesis that bring about differential responses related to regeneration potential” is too general. Move in the Discussion section or remove it.
Authors’ response: The whole text as seen above was too general thus we decided to be fully removed from the Conclusion section in an attempt to short the extent of the whole text as well.
Reviewer: The phrase “A suitable in vitro adventitious root culture system in liquid media at 250 mL Erlenmeyer flasks on a continuous rotary shaker derived from callus cultures initiated from leaf-, petiole- and root-explants in different composition agar-solidified media under different incubation periods for the production of important bioactive molecules, such as rosmarinic acid and further optimization of the methodology for scale up production in a 2 L Balloon Bubble Type Bioreactor (BBTB) system is underway” remove it, is not necessary to be inserted. I said that just because I wish in the future to see this article published.
Authors’ response: In the 2nd revised manuscript, this sentence does not exists, it was removed.

Reviewer 2 Report
Table 1-2 Difference analysis process can be deleted;
All the letters represented by the figures are optimal placed in the figure, and each figure must have a bar.
Author Response
Journal: Plants (ISSN 2223-7747)
Manuscript ID: plants-2106767
Type: Article
Title: Cretan dittany (Origanum dictamnus L.), a valuable local endemic plant: In vitro regeneration potential of different type of explants for conservation and sustainable exploitation
Authors: Virginia Saropoulou, Eleni Maloupa, Katerina Grigoriadou*
Section: Plant Genetic Resources
Special Issue: Plant Genetic Resources In Vitro Culture: Present Situation and Prospects for Propagation, Conservation and Sustainable Use
Response to Reviewer #2 comments
Review Report Form
|
(x) English language and style are fine/minor spell check required |
|
Yes |
Can be improved |
Must be improved |
Not applicable |
|
|
Does the introduction provide sufficient background and include all relevant references? |
( ) |
(x) |
( ) |
( ) |
|
Are all the cited references relevant to the research? |
( ) |
(x) |
( ) |
( ) |
|
Is the research design appropriate? |
( ) |
(x) |
( ) |
( ) |
|
Are the methods adequately described? |
( ) |
(x) |
( ) |
( ) |
|
Are the results clearly presented? |
( ) |
(x) |
( ) |
( ) |
|
Are the conclusions supported by the results? |
( ) |
(x) |
( ) |
( ) |
Comments and Suggestions for Authors
Reviewer: Table 1-2 Difference analysis process can be deleted;
Authors’ response: In the 2nd revised manuscript, we revised the text describing the process followed for difference analysis in section 4. Materials and Methods and sub-section 4.4. Statistical analysis. In this framework, p-values derived from 2-way ANOVA or 3-way ANOVA and General Linear Model for determining the effect of main and combined factors and their between interactions were deleted. Therefore, the same and only one-way ANOVA pattern was followed and described in all experiments regarding data in both Tables and Figure graphs, for consistency reasons. In additions, all footnotes related to statistical difference below Tables (Table 1, Table 2) and caption below 2c figure graph were also deleted. A general description is provided only as text in 4.4. Statistical analysis section. See page 12 (lines 27-41) & page 12 (lines 1-13).
Reviewer: All the letters represented by the figures are optimal placed in the figure, and each figure must have a bar.
Authors’ response: In figures representing results data in the form of graphs, different bars per graph denoted by different letters indicate statistically significant differences among treatments (p ≤ 0.05) and error bars in each bar per graph are standard errors. For each figure graph, a horizontal scale bar on X-axis is provided in its respective caption in parenthesis (Figure 1a-1h, Figure 2c). For example: graph orizontal scale bar on X-axis: 1 unit = 10% ex vitro survival rate or 1 unit = 0.5 cm root length. Furthermore, in figures representing photo images and not graphs, a scale bar of 1 cm is provided on each multi-plate image (Figure 2a, 2b, 40 2d, Figure 3a-3h, Figure 4a-4g).
